# Experimental Investigation of Light Steel Framing Walls under Horizontal Loading

**Dalila M. Lopes** [1,†]**, António P. C. Duarte** [2,*] **and Nuno Silvestre** [3]

1   School of Business, Engineering and Aeronautics, Instituto Superior de Educação e Ciências–ISEC Lisboa, Alameda das Linhas de Torres 179, 1750-142 Lisboa, Portugal
2   CERIS, Department of Civil Engineering, Architecture and Georesources, Instituto Superior Técnico, University of Lisbon, Av. Rovisco Pais 1, 1049-001 Lisboa, Portugal
3   IDMEC, Department of Mechanical Engineering, Instituto Superior Técnico, University of Lisbon, Av. Rovisco Pais 1, 1049-001 Lisboa, Portugal
*   Correspondence: antonio.duarte@tecnico.ulisboa.pt
†   Dedicated to the memory of Dalila M. Lopes.

**Abstract:** The mechanical behavior of light steel framing (LSF) walls under horizontal (shear) loadings is reported and assessed in this paper. In total, an experimental program with twelve LSF walls (six under monotonic and six under cyclic loading) was conducted, and the main parameters investigated were (i) the thickness and (ii) the material used as the cladding (OSB, a plasterboard, and a steel sheet), (iii) the spacing between fasteners (150 or 75 mm), and (iv) the influence of using steel bracing elements. It is concluded that doubling the number of fasteners and increasing the thickness of OSB by 80% lead to increases in ultimate loads, respectively, of 33 and 13%. The ductility index of the walls with steel sheets was 50 to 75% lower than those of the remaining walls. The wall with the steel strap x-bracing system presented (i) the lowest initial rigidity (a diaphragm effect could not be triggered with these elements) and (ii) the highest damage extent at the end of testing (a damage parameter of 0.85, due to damage of the steel strap-to-steel structure connection). It is confirmed that the results obtained with testing of the walls under a monotonic load can be good predictors of their behavior under cyclic loading as, for instance, the ultimate loads of walls under both loading cases present an average difference of 4%.

**Keywords:** light steel framing; horizontal loads; experimental investigation; monotonic and cyclic loading; ductility; damage

## 1. Introduction

Light steel framing (LSF) structures, made of cold-formed steel members, have been gaining interest and relevance in the construction of residential houses and small buildings over the past three decades. In this context, the aim of LSF walls is two-fold: (i) to assure the transmission of vertical loads to the foundations and (ii) to act as diaphragms, and resist horizontal actions stemming from either wind or earthquakes. Between 1993 and 2002, the number of LSF-built houses per year increased 25 times [1] in the United States of America; in Europe, especially in the United Kingdom and several northern European countries, the adoption of LSF solutions has also increased significantly over the past years [2].

Associated with the increasing adoption of LSF-built houses, it was not surprising to observe a growth of research studies in this specific area, as several experimental [3–20] and numerical [14,19,21,22] studies were developed to investigate the mechanical behavior of LSF structures. Recently, Hasanali et al. [23] published a critical review of the developments and challenges related to the behavior of LSF seismic-resistant systems. These authors pointed out as main directions of future research (i) the adoption of comprehensive measures to prevent non-ductile failure of LSF structural systems under in-plane shear load, (ii) the investigation of structures that resist a combination of vertical and horizontal

loads, as well as of fire, and (iii) the investigation of the structural behavior of multi-story buildings for seismic areas. In fact, the latter was a topic experimentally studied by Zhou et al. [24], who developed an experimental campaign to assess the seismic response of a six-story building using LSF.

As for previous studies on the behavior of LSF walls, by the end of the 20th century, Serrette and Ogunfunmi [3] studied the behavior of square LSF shear walls (2.44 m by 2.44 m) considering three shear-resistant systems: (i) a steel strap x-bracing system applied in one of the faces of the walls, (ii) plasterboard cladding in one of the faces of the walls, and (iii) plasterboard cladding in both faces of the walls together with a steel strap x-bracing system in one of the faces. The authors concluded that the behavior of the wall with the first shear-resistant system was governed by the yielding of the x-bracing system, whereas the behavior of the walls with plasterboards was governed by cracking of the cladding near the edges (to a lesser extent when the x-bracing system was also used). In 2004, Tian et al. [5] experimentally studied the behavior of six rectangular LSF walls (2.45 m by 1.25 m) under monotonic lateral loading, which consisted of an outer frame and an interior stud. These authors studied the influence of using OSB (oriented strand board) and CPB (cement particle board) as cladding and of using (i) a single steel strap x-bracing system (full width of the wall) in (i.a) one or (i.b) two sides of the walls and (ii) two x-bracings systems (half-width of the wall) in both sides of the walls. The authors concluded that for frames sheathed with OSB or CPB, failure occurred due to shear tearing of the cladding by the fasteners (pull-through) and that either (i) the increase in board thickness or (ii) the decrease in fastener spacing leads to an increase in the wall strength. Additionally, the use of a single x-bracing system (full width of the wall) compared to the use of two (half-width of the wall) was verified to be more efficient. Later, Al-Kharat and Rogers [6] experimentally studied the behavior of a total of sixteen squared LSF walls (2.44 m by 2.44 m) braced with steel straps subjected to both monotonic and cyclic loadings, focusing their investigation on the influence of the dimensions of these reinforcing elements on the ductility of the walls. Straps with combinations of (thickness; width) of (i) (1.22; 58.4 mm), (ii) (1.52; 101 mm), and (iii) (1.91; 152 mm) were studied. Generally, the authors concluded that the walls with straps with the smallest cross-sections were the only ones that contributed to a considerable ductility of the walls, whereas the remaining did not attain the estimated load required to trigger yielding and, therefore, the collapse of the walls was triggered by the failure of some of the remaining elements (gusset plates and tracks) near the corners of the walls. Dubina [7] conducted an experimental study on the behavior of LSF walls under horizontal actions, in which six series of walls with typical constructive solutions and elements (e.g., OSB panels, plasterboards, bracing, corrugated steel sheet, and the existence of doors) were tested under both monotonic and cyclic loadings. This author concluded that the ultimate strengths of the walls under cyclic loading were about 10% lower than those of the walls tested under monotonic loading. The use of the plasterboard together with a corrugated sheet slightly improved ductility and improved the ultimate strength of the walls by circa 17%. This author also performed tests on the fasteners and developed finite element models of the tested walls. Additionally, in the scope of the influence of constructive elements, such as doors and windows, on the behavior of walls, more recently, Yan et al. [8] studied the mechanical behavior of LSF walls with high aspect ratios under monotonic lateral loading. Moghimi and Ronagh [9] focused their investigation on the improvement of connection detailing of strap-braced LSF walls to be used in seismic regions and studied, among other solutions, (i) the addition of brackets at the corners of the walls to improve their lateral performance by, for instance, reducing the steel strap buckling length and (ii) the use of perforated straps to help trigger yielding and to prevent brittle failure. Xiang et al. [10] and Yu et al. [11] also focused their investigations on the influence of bracing systems on the seismic behavior of LSF walls by studying, respectively, the use of X- and K-strap bracing systems and of a special energy dissipation bracing to improve the ductility of the corrugated sheet sheathed shear walls. Alternatively, to improve the mechanical behavior of LSF walls by means of the end studs instead of by means of bracing, sheathing, or

cladding, recently, the use of reinforced end studs [15], the consideration of different cross-sections for studs [16], and the adoption of concrete filled steel tubes [17] have also been addressed by some authors.

In recent years, in addition to the investigation of the mechanical behavior of LSF walls under monotonic and cyclic loadings, as presented before, experimental and numerical investigations on their thermal behavior [25,26], due to the high difference in thermal performances between the (steel) wall structure and cladding materials, and their fire behavior [27,28], due to the high sensitivity of steel mechanical properties to high temperatures and fire, have also drawn the attention of several authors. Within the scope of the thermal behavior of the walls but also accounting for the sustainability of the construction sector and the preservation of the lightweight of these structures, several authors [18–20] studied the use of cladding in walls with non-conventional materials, which are by-products of other industries. Paper straw board, flue gas desulfurization gypsum, and a composite made of straw bales, welded wires, and cement mortars are among some of the solutions investigated.

Nevertheless, and despite the relevance of cladding and bracing on the mechanical behavior of LSF walls under horizontal loads, as shown above, most design codes, such as the EN 1993-1-3, still lack design methodologies to deal with this subject. Bearing this in mind, in 2004, Chen [29] (cited by Yanagi [30]) developed an elastic simplified strength model for LSF walls sheathed by wood-based materials under in-plane shear loading. The main assumptions adopted by this design model were that (i) the tracks and studs remain rigid during loading and were hinged at the corners of the walls, (ii) the wood-based panels were also rigid, (iii) the work performed by the loading was totally dissipated by the connections between the sheathing and the frame, and (iv) the load–displacement curve of the connections was idealized as bi-linear elastic-plastic. In 2009, Fiorino et al. [31] developed a design procedure for the seismic design of LSF walls, which is based on three nomographs: (i) one linear dynamic, (ii) one nonlinear static, and (iii) one capacity design nomograph. The proposed design approach is applicable to LSF walls with wood-based panels or plasterboard used as cladding; walls with openings cannot be designed according to the proposed methodology and the assumed failure mode is through sheathing–fastener failure. More recently, Yanagi and Yu [32] proposed an effective strip method for the design of LSF walls with steel sheathing under in-plane shear loading. The method assumes that the inclined tensile loading that arises on the sheathing due to shear loading of the wall is mainly concentrated in an effective strip; this allows the design of these walls to be treated as if they were reinforced with a steel strap x-bracing system. Then, the strength of a given wall is assumed to be the lowest value between (i) the shear strength of the sheathing–frame connections or (ii) the yield stress of the sheathing multiplied by its thickness and the calculated effective strip width.

In view of the aforementioned studies, the main objective of this paper is to contribute to the existing knowledge and the development of future design provisions for LSF walls under in-plane shear loading by presenting an experimental investigation on the influence of cladding and bracing on their mechanical behavior. In particular, the objective is to identify the influence of (i) the material and (ii) thickness of the cladding, and (iii) fastener spacing on the mechanical behavior of the walls. For this, firstly, the experimental program conducted is presented in detail. Then, the experimental results are shown and thoroughly discussed; the results include the presentation of load vs. displacement curves, ultimate loads, and failure modes, as well as the investigation of the ductility and damage of the walls. Throughout the presentation of results, the main factors affecting the properties of the walls are identified and discussed. Finally, the main conclusions of the work are drawn.

## 2. Experimental Program

### 2.1. Specimens and Materials

The experimental program comprised the testing of 12 walls subjected to in-plane lateral (shear) loading: 6 walls were subjected to monotonic loading and 6 walls were

subjected to cyclic loading. The reference wall had a 12 mm thick OSB cladding on one of the sides and the fastener spacing (along the wall contour and central stud) was 150 mm. The parameters varied in the remaining walls were:

- a fastener spacing of 75 mm (instead of 150 mm);
- the use of a plasterboard cladding (with a thickness of 13 mm) in addition to the 12 mm thick OSB cladding;
- OSB thickness of 22 mm (instead of 12 mm);
- the use of a 1.5 mm thick steel strap x-bracing system;
- the use of a 1.5 mm thick steel sheet.

Table 1 provides an overview of the main characteristics of the tested walls and the adopted corresponding labeling. Take, as an example the wall labeled as 'O12_150_M', (i) 'O' stands for OSB ('P' for plasterboard, 'XB' for x-bracing, and 'SS' for steel sheet), (ii) '12' (or the value succeeding the previous designation) stands for the thickness, in millimeters, of the cladding or bracing, (iii) '150' (or '75') stands for the spacing, in millimeters, of fasteners, and (iv) 'M' (or 'C') informs that the wall was subjected to monotonic (or cyclic) loading.

**Table 1.** Details and labeling of the tested walls.

| Specimen | Cladding/ Bracing | Schematic Drawing |
|---|---|---|
| O12_150_M O12_150_C | 12 mm thick OSB | |
| O12_75_M O12_75_C | 12 mm thick OSB | |
| O22_150_M O22_150_C | 22 mm thick OSB | |
| SS1.5_150_M SS1.5_150_C | 2050 × 1200 × 1.5 mm steel sheet | |
| O12_P13_150_M O12_P13_150_C | 12 mm thick OSB + 13 mm thick plasterboard | |
| XB1.5_150_M XB1.5_150_C | 2200 × 100 × 1.5 mm steel straps | |

The height and width of the tested walls are, respectively, H = 2050 mm and B = 1200 mm. The main frame of the walls, steel straps, and steel sheets were made of S280 GD cold-formed steel elements. The top and bottom tracks were U93/1.5, whereas the three (one central and one at each side) studs were C90/1.5. The connections between (i) the tracks and studs (main frame elements), (ii) the x-bracing elements and main frame elements, and (iii) the steel sheet and main frame elements were ensured by Φ4.2 × 25 self-drilling fasteners. The connections between the OSB with 12 or 22 mm and the main frame elements were accomplished, respectively, by Φ4.8 × 32 and Φ5.2 × 55 self-drilling fasteners. Finally, the connections between the plasterboard and the main frame elements were ensured by

Φ3.5 × 25 self-drilling fasteners. Table 2 presents the mechanical properties of the materials used in the tested LSF walls.

**Table 2.** Nominal values of the mechanical properties of materials.

| | $E_{longitudinal}$ (GPa) | $E_{transversal}$ (GPa) | $f_{u,bending}$ (Longitudinal) (MPa) | $f_{u,bending}$ (Transversal) (MPa) |
|---|---|---|---|---|
| OSB3 | 3.5 | 1.4 | 18.0 | 9.0 |
| Plasterboard | - | - | 6.2 | 2.4 |
| | E (GPa) | $f_y$ (MPa) | $f_u$ (MPa) | $\varepsilon_u$ (-) |
| S280 GD | 210 | 280 | 360 | 0.18 |

Note: E—modulus of elasticity; $f_y$—yield strength; $f_u$—ultimate strength; $\varepsilon_u$—ultimate strain.

### 2.2. Test Setup and Instrumentation

Figure 1a shows a schematic drawing of the setup adopted in the testing of the walls. As can be seen, it mainly comprised (i) a steel frame, (ii.a) a *Dartec* hydraulic reversible actuator (capacity of 250 kN and a maximum stroke of 400 mm), (ii.b) a *TML* load cell (capacity of 300 kN), (iii) a *TML* wire transducer (with a maximum stroke of 500 mm), (iv) two steel sub-structures, suspended from the top beam of the frame which prevented the top tracks of the walls from presenting (sway mode) out-of-plane displacements (Figure 1b), and (v) an Ω cross-section shaped beam that allowed clamping the LSF walls to the steel frame bottom beam (Figure 1c). Additionally, two *dywidag* bars were used together with the actuator to allow transferring the load to the opposite side of the LSF walls, in case of reverse (cyclic) loading (Figure 1b).

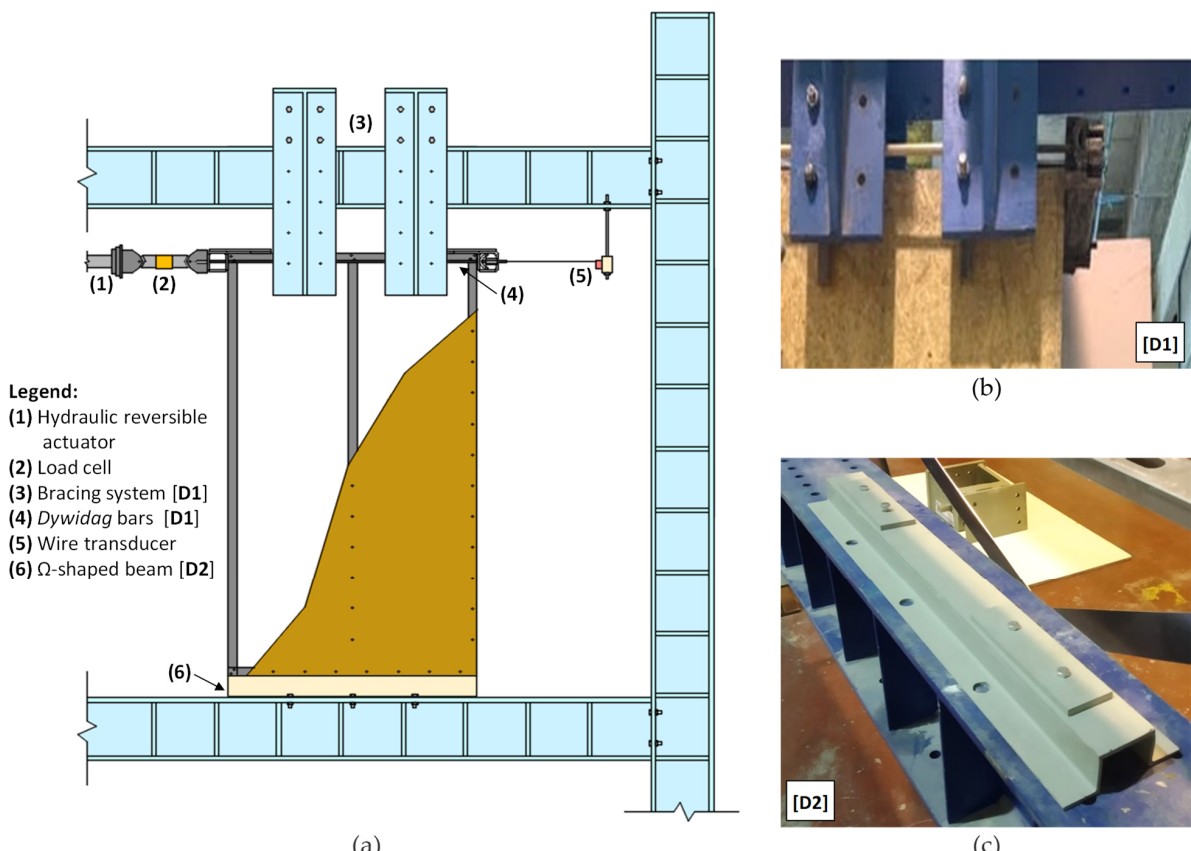

Legend:
**(1)** Hydraulic reversible actuator
**(2)** Load cell
**(3)** Bracing system **[D1]**
**(4)** *Dywidag* bars **[D1]**
**(5)** Wire transducer
**(6)** Ω-shaped beam **[D2]**

**Figure 1.** (**a**) Schematic drawing of the test setup and details (**b**) D1 and (**c**) D2.

In addition to monitoring the horizontal displacement (by means of the wire displacement transducer), the walls with x-bracing and steel sheeting were also instrumented (each) with two *TML* electric strain gauges. Their location within the walls is shown ahead in Section 3.

### 2.3. Loading Protocols and Experimental Procedure

The monotonic tests were performed under displacement control imposed by the actuator, at a rate of 0.2 mm/s, until the load decreased to a value of about 50% of the maximum load of the tested wall.

The adopted cyclic loading protocol was the one recommended by the ECCS [33], in which the increasing amplitude of the cyclic horizontal displacement, d, depends on the prior knowledge of a yield displacement (obtained with the monotonic tests), $d_y$, according to (Figure 2):

$$d = \pm\frac{d_y}{4}; \pm\frac{d_y}{2}; \pm\frac{3d_y}{4}; \pm d_y; \pm 2d_y; \pm 2d_y; \pm 2d_y; \pm 4d_y; \pm 4d_y; \pm 4d_y; (\dots) \tag{1}$$

where the values of $d_y$ were estimated based on the monotonic test results as those corresponding to the end of the elastic range. The method recommended by the ECCS [33] to estimate the values of $d_y$ was considered, and the value of $d_y$ is given by the abscissa of the intersection between a line that crosses the origin and has a slope defined by the initial rigidity, and one that contains the point of ultimate load and has a slope corresponding to 10% of the initial rigidity (illustrated in Figure 3a). The load application in the cyclic tests was also based on displacement control, and the same rate as that considered for the monotonic tests was adopted (0.2 mm/s).

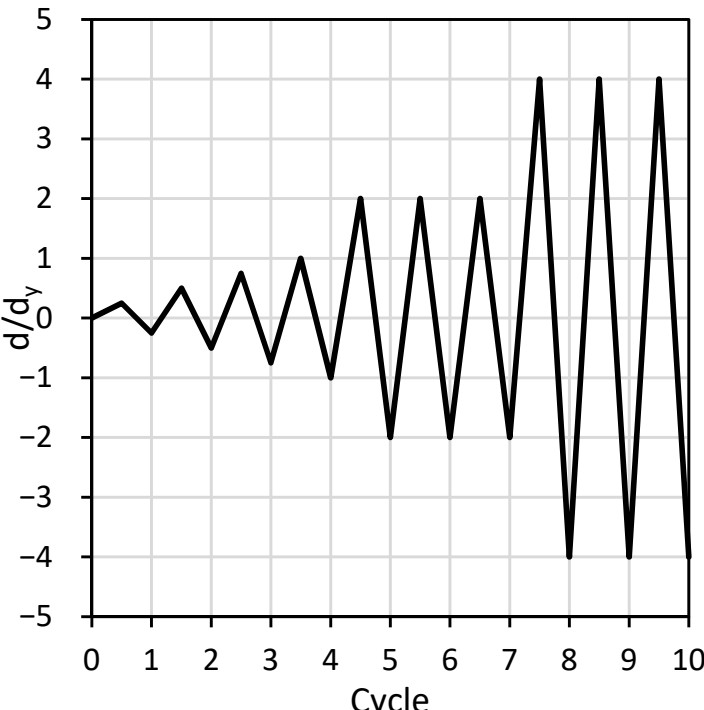

**Figure 2.** Cyclic loading protocol according to the ECCS [33].

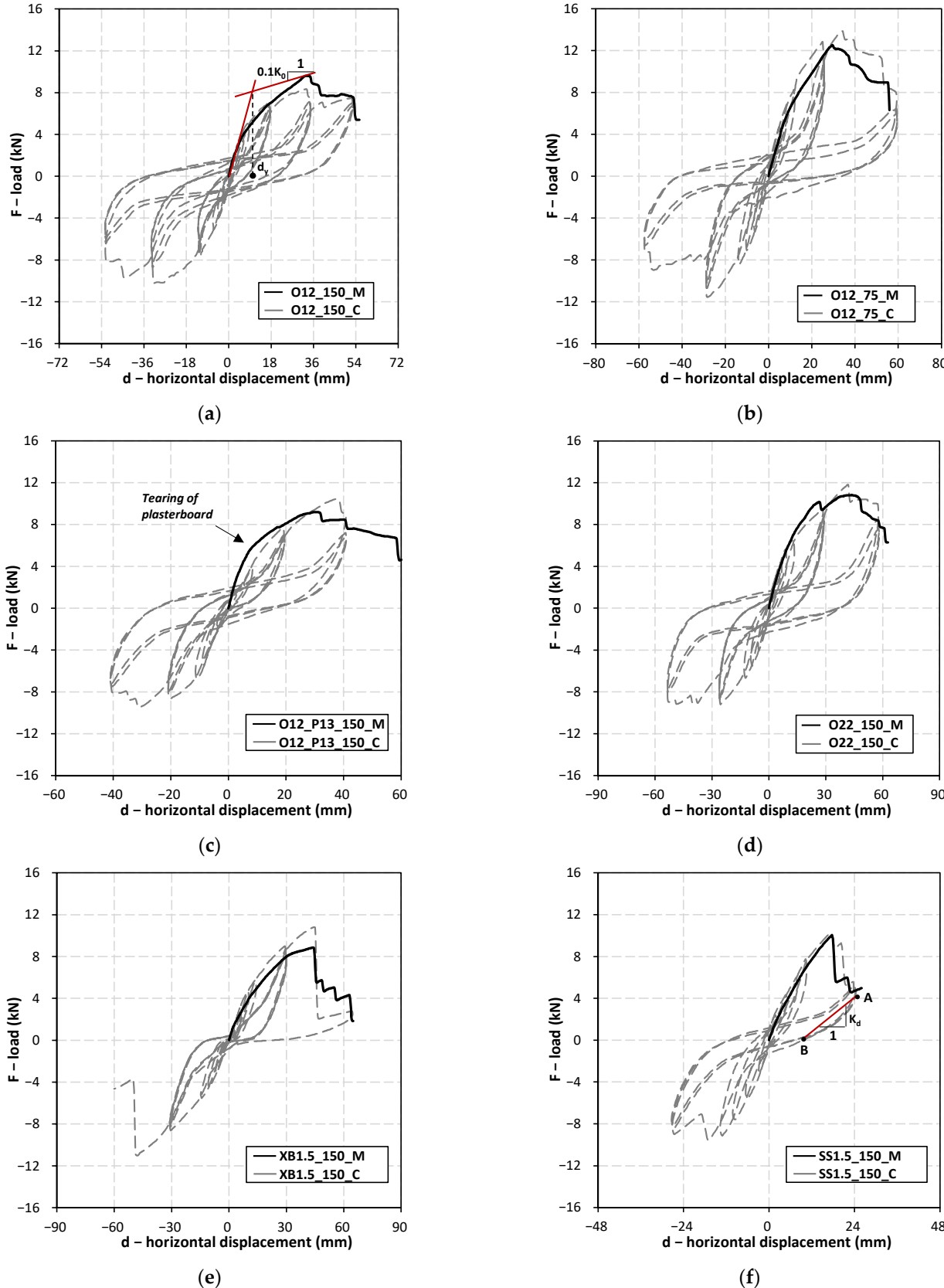

**Figure 3.** Load vs. horizontal displacement of the tested walls: (**a**) O12_150_M and O12_150_C, (**b**) O12_75_M and O12_75_C, (**c**) O12_P13_150_M and O12_P13_150_C, (**d**) O22_150_M and O22_150_C, (**e**) XB1.5_150_M and XB_150_C, and (**f**) SS1.5_150_M and SS1.5_150_C.

## 3. Results and Discussion

### 3.1. Load–Displacement Curves, Ultimate Loads, and Failure Mechanisms

Table 3 shows the main results obtained with the testing of the walls, in particular, (i) initial rigidity ($K_0$), (ii) yield displacement ($d_y$), (iii) ultimate load ($F_u$), (iv) ductility index ($\mu$) (addressed ahead), and (v) identification of the overall failure mechanism. For the walls subjected to cyclic loading, the average of the maximum loads of the positive and negative half-cycles is shown. Additionally, Figure 3 depicts the load (F) vs. horizontal displacement (d) of the walls tested under monotonic and cyclic loading.

**Table 3.** Experimental results obtained with testing of LSF walls.

| Specimen | $K_0$ (kN/mm) | $d_y$ (mm) | $F_u$ (kN) | $\mu$ (-) | Overall Failure Mechanism |
|---|---|---|---|---|---|
| O12_150_M | 0.66 | 9.0 | 9.6 | 5.8 | Tearing of OSB |
| O12_75_M | 0.72 | 15.0 | 12.5 | 2.9 | Tearing of OSB |
| O12_P13_150_M | 0.92 | 10.0 | 9.2 | 4.8 | Tearing of OSB |
| O22_150_M | 0.65 | 16.0 | 10.8 | 3.4 | Tearing of OSB |
| XB1.5_150_M | 0.39 | 16.0 | 8.9 | 2.8 | Shear cut and pull out of fasteners |
| SS1.5_150_M | 0.74 | 13.0 | 10.0 | 1.4 | Shear cut and pull out of fasteners |
| O12_150_C | 0.66 [a] | 9.0 [a] | 9.3 | - | Tearing of OSB |
| O12_75_C | 0.72 [a] | 15.0 [a] | 12.7 | - | Tearing of OSB |
| O12_P13_150_C | 0.92 [a] | 10.0 [a] | 10.0 | - | Tearing of OSB |
| O22_150_C | 0.65 [a] | 16.0 [a] | 10.5 | - | Tearing of OSB |
| XB1.5_150_C | 0.39 [a] | 16.0 [a] | 10.9 | - | Shear cut and pull out of fasteners |
| SS1.5_150_C | 0.74 [a] | 13.0 [a] | 9.9 | - | Shear cut and pull out of fasteners |

[a]—Based on/assumed to be the same as that of the walls tested under monotonic loading.

The first observation of Table 3 and Figure 3 shows that the monotonic and (envelopes of the) cyclic F-d curves of the walls are qualitatively and quantitatively similar. They exhibit (i) an approximately linear initial branch, in which all materials and connections present linear elastic behaviors, (ii) followed by a second non-linear branch, with lower rigidity than the previous branch, which is mainly due to local buckling and yielding of the tracks and studs near the corners of the walls (Figure 4a) (iii) up to the ultimate load, $F_u$, of the walls, after which (iv) the load decreases with the increase in displacement. Unlike the results presented by Dubina [7], which reported that the ultimate load of the walls under cyclic loading is about 10% lower than that of their counterparts tested under monotonic loading, the average ratio between maximum loads of walls tested under cyclic and monotonic loading in this work is of $1.04 \pm 0.09$. From the obtained results, it seems that the strength of the walls under cyclic events can be predicted with fair accuracy by tests under monotonic loading.

Concerning the overall failure mechanisms that governed the failure of the walls, two groups can be perceived: (i) in the first (composed by walls O12 _150_M and _C, O12 _75_M and _C, O12_P13_150_M and _C, and O22 _150_M and _C) failure was governed by shear tearing of the OSB (Figure 4b) (in the case of walls O12_P13_150_M and C, the plasterboard was also torn by fasteners but this did not lead to failure of the walls; therefore, in this work, it is addressed as a partial failure mechanism), whereas (ii) in the second group (composed by walls XB1.5_150_M and _C, and SS1.5_150_M and _C) overall failure was governed by the shear cut and pull out of fasteners (Figure 4c). Generally, the former group presents a less brittle behavior than the latter, as denoted by F-d curves in Figure 3 (and discussed ahead in more detail).

With respect to the rigidity of the first group of walls, Table 3 indicates that (i) the initial rigidity increases by about 10% when the distance between fasteners decreases by 50% (comparison of walls O12_150_M and O12_75_M), (ii) the rigidity of the walls increases 40%

when the plasterboard is also used together with the OSB (comparison of walls O12_150_M and O12_P13_150_M), and (iii) the rigidity remains unaltered when the thickness of the OSB increases about 80% (comparison of walls O12_150_M and O22_150_M). Given the aforementioned, it becomes evident that the rigidity of the walls (in this first group) is influenced not only by the thickness and stiffness of the cladding, but also by the number of fasteners. The fasteners act as springs and have a given (finite) rigidity. Hence, the higher the number the more effective the motion compatibility (and the lowest the slippage) between the steel structure and the cladding, the more effective the diaphragm effect. Notice that the wall with the plasterboard has both a higher overall cladding thickness and the number of fasteners (although distributed in two planes).

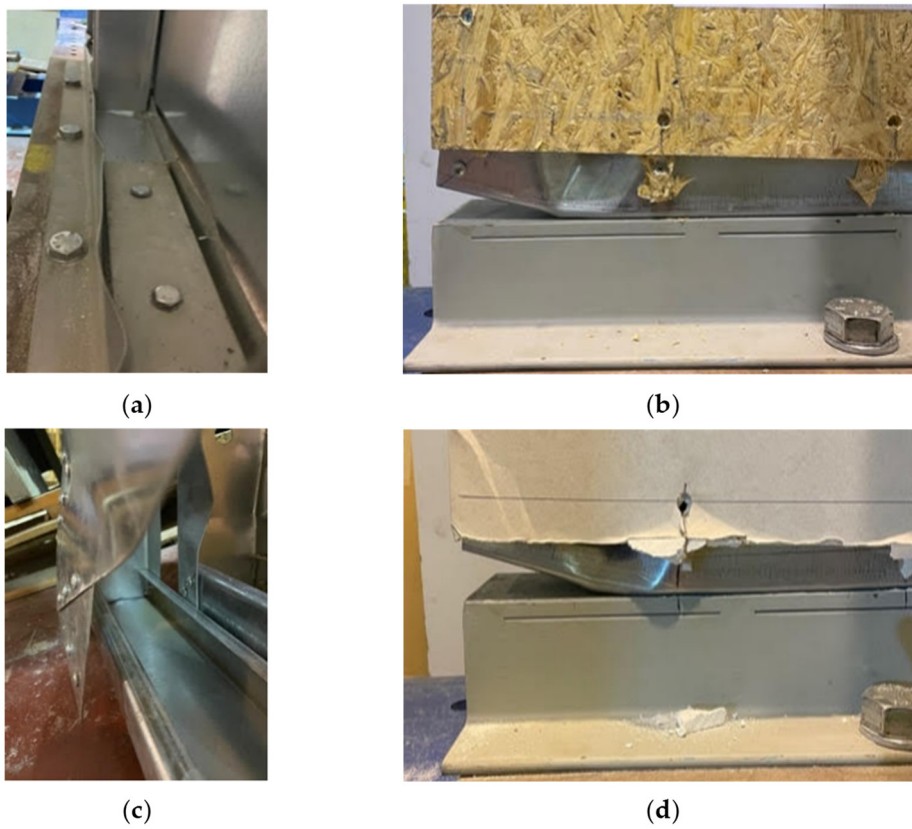

(a)  (b)

(c)  (d)

**Figure 4.** (**a**) Local buckling and yielding of the bottom track (partial failure mechanism), (**b**) tearing of the OSB (overall failure mechanism), (**c**) shear cut and pull out of fasteners (overall failure mechanism), and (**d**) tearing of the plasterboard (partial failure mechanism).

As for the ultimate loads of the first group of walls, Table 3 shows that (i) the decrease in the distance between the fasteners in 50% (comparison of walls O12_150_M and _C and O12_75_M and _C) leads to an increase of about 33% of the ultimate load of the walls, (ii) an increase in thickness of the OSB of about 80% (comparison of walls O12_150_M and _C and O22_150_M and _C) leads to an increase of circa 13% of the ultimate load, and (iii) the use of the plasterboard leads, on average, to invariance of the ultimate load of the walls. Indeed, during the testing of the walls, the plasterboards failed (torn by the fasteners, Figure 4d) earlier than the OSB but only the latter led to the failure of the walls, whereas the former only led to a rigidity decrease in the walls. Table 2 shows that the strengths of the plasterboard are about three times lower than those of the OSB. This earlier tearing of the plasterboard is also visible by a higher decrease in rigidity of the F-d curves of walls with this element (presented in Figure 3c) compared to those of the remaining (O12 _150_M and C—Figure 3a, O12 _75_M and _C—Figure 3b, and O22 _150_M and _C—Figure 3d). Given the previous remarks, it can be concluded that for the first group of walls, the ultimate load is dictated by the number of fasteners and the strength and thickness of the OSB.

For the second group of walls (XB1.5_150_M and _C, and SS1.5_150_M and _C), as presented in Table 3, failure was mainly governed by shear cut and pull out of fasteners in the corner regions of the walls. The rigidity of wall XB1.5_150_M is the lowest of all the walls tested, as there is no diaphragm effect, and the rigidity depends essentially on that of the diagonal element in tension (that in compression buckled for small displacement magnitudes). The rigidity of wall SS1.5_150_M is within the range of all the remaining and both walls of this second group presented ultimate loads within the range of those with non-metallic claddings (with the exception of O12 _75_M and _C, which present markedly higher ultimate loads).

### 3.2. Ductility, Buckling, and Damage

For both walls in the second group, it was not possible to take advantage of the ductility of the reinforcing elements (the steel bracing and steel sheet), as denoted by the sharp decreases in the load-carrying capacity of the walls after their ultimate load was attained (Figure 3e,f). As mentioned previously, the reinforcing elements of these walls were monitored with electric strain gauges (one placed at the front and one placed at the back of the steel elements) and the results obtained (strain vs. load) for the monotonic tests can be observed in Figure 5a,b. In both cases, it can be seen that the maximum strains monitored, respectively, of circa $\varepsilon$ = 320 µstrain and $\varepsilon$ = −60 µstrain for walls XB1.5_150_M and SS1.5_150_M, are much lower than that associated with yielding ($\varepsilon_y$) of the steel elements $\varepsilon_y \approx 280/210{,}000 \times 1 \times 10^6 = 1333$ µstrain. This means that these elements were still in the elastic regime when the respective walls failed. Figure 5a also shows that after wall XB1.5_150_M reached its ultimate load, the average strain of the steel strap became null, proving that all the fasteners failed at one end (as shown in Figure 4c). It can be inferred that the steel strap was oversized with respect to the connections at the corners of the wall if its function was to confer ductility to the wall. To take advantage of the ductility of this element, a smaller width or a stress raiser (e.g., a hole) [9] would have had to be considered.

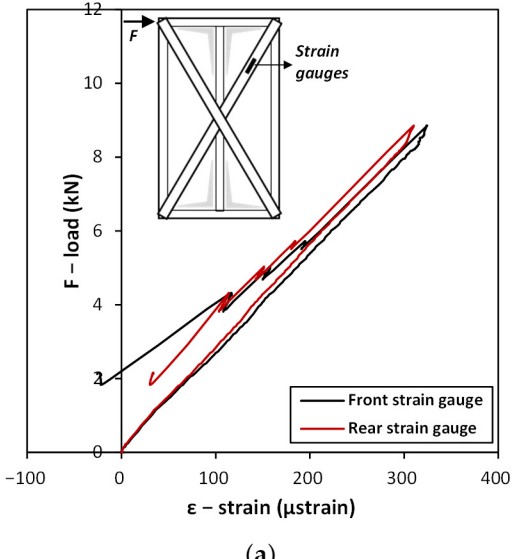

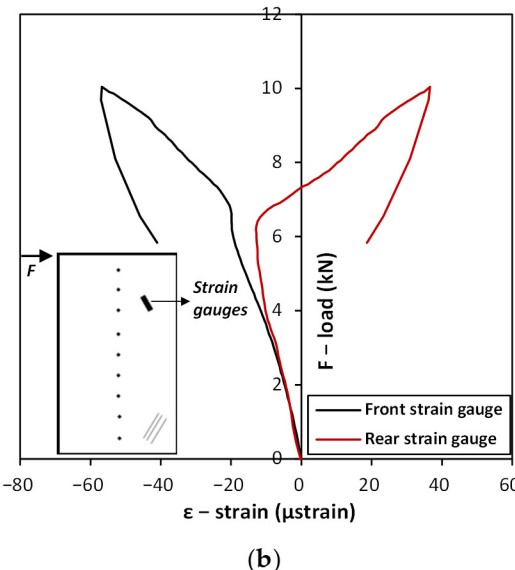

(a)

(b)

**Figure 5.** Strain gauges measurements in walls (**a**) XB1.5_150_M and (**b**) SS1.5_150_M.

Regarding wall SS1.5_150_M, the strain did not become null after the wall reached its ultimate load since the steel plate still had the ability to transfer load throughout the tracks and studs after the fasteners at the corners failed. It is also interesting to observe, in Figure 5b, that the strain gauges located at the front and back of the steel sheet were both initially monitoring compressive strains (these were positioned perpendicular to the halve waves that were expected to form if the steel sheet buckled), as expected, but then

started deviating from one another. Indeed, when this occurred, shear buckling of the steel sheet took place which led the strain gauge on the concave side to continue monitoring compressive strains and led the strain gauge on the convex side to shift from compressive to tensile strain. Furthermore, after buckling, the rate of increase in strain with the increase in load became higher than before, Figure 5b. The buckling load, based on Figure 5b, is approximately $F_{cr} \approx 6.5$ kN, and it can be seen that the post-buckling trajectory is stable. For comparison purposes, an estimate of the buckling load of the steel sheet can be made considering two steel plates (with half the width of the wall due to the existence of the central stud) that buckle simultaneously and are simply supported along all edges (tracks, edge studs and central stud). It can be obtained using the well-known equation for local buckling of plates [34]:

$$\tau_{cr} = K_\tau \frac{\pi^2 E}{12(1-\nu^2)} \left( \frac{t^2}{B_{plate}^2} \right) \tag{2}$$

where

$$K_\tau = 5.35 + \frac{4}{\left( H/B_{plate} \right)^2} \tag{3}$$

and t is the thickness of the steel sheet (t = 1.5 mm), $\nu$ is Poisson's coefficient of steel ($\nu$ = 0.30), E is Young's modulus of steel (E = 210 GPa), H is the height of the wall (H = 2050 mm), and $B_{plate}$ = 0.5B = 600 mm is the width of the plate considered for the calculation of the buckling load. An estimated buckling load of $F_{cr,est} = \tau_{cr} \cdot t \cdot B$ = 12.1 kN is obtained, which is about 85% higher than the experimental value. This difference can be explained by the fact that the studs do not have an infinite bending stiffness and, therefore, the steel sheet is not exactly simply supported. Indeed, it is over deformable supports and, therefore, when the steel sheet buckles, the studs present out-of-plane deflections.

Besides the evaluation of rigidity, ultimate load, and failure modes of the walls, their ability to dissipate energy is also of utmost relevance. Hence, a ductility factor ($\mu$) was estimated for the walls tested under monotonic loading [6]:

$$\mu = \frac{d_{0.8F_u}}{d_y} \tag{4}$$

where $d_{0.8F_u}$ is the displacement that corresponds to a load that is 80% of the wall's ultimate load in the post-peak branch of the F-d curve and $d_y$ is the yield displacement of the wall.

Table 3 presents the ductility values of the walls tested under monotonic loading. As can be seen, the less brittle wall is O12_150_M with a $\mu$ = 5.8, whereas the most brittle is wall SS1.5_150_M with a $\mu$ = 1.4, four times lower than that of the former. It can also be concluded that decreasing the fasteners spacing by 50% leads to a decrease in the ductility factor by about 50%, whereas increasing the OSB thickness by 80% decreases the ductility factor by about 40%. The use of plasterboard or the x-bracing system also leads to decreases in ductility, respectively, of 20 and 50%.

In addition to evaluating the ability of the walls to dissipate energy, it is also relevant to verify if such energy is being dissipated by means of plasticity or by means of damage. During plastic deformation, the unloading of a given structure is elastic; however, when damage occurs, unloading follows a damaged elasticity path (of lower rigidity). Hence, to evaluate the aforementioned, a damage parameter (D) was determined for the walls under cyclic loading, which assumes the usual form:

$$D = 1 - \frac{K_d}{K_0} \tag{5}$$

where $K_d$ is the damaged rigidity of a given wall upon unloading and $K_0$ is its initial rigidity (Table 3). Because the unloading path of the walls assumes some nonlinear branches, to consistently estimate the damaged rigidity, a secant damage rigidity was assessed by evaluating the slope of a line connecting two points: A and B, with the coordinates

A[maximum displacement of a given half-cycle; corresponding load] and B[null load; corresponding displacement]; the procedure is illustrated in Figure 3f. The following was also considered: (i) the cycles up to $d_y$ were not considered, assuming that no yielding nor damage had occurred up to that point, and (ii) for the sets of three cycles with a given amplitude (multiples of $d_y$, as per Equation (1)), only the first cycle was evaluated, as it can be seen (Figure 3) that the unloading rigidity of the remaining cycles is similar to that of the first.

Figure 6 presents the evolution of the damage parameter, D, with the increase in the horizontal displacement, d, at the end of each (first) cycle after $d_y$. The value of D is the average of those determined for each (positive and negative) half-cycle (an exception is wall XB1.5_150_C, for which the test ended before the unloading of the negative half-cycle with an amplitude of $-4d_y$ was attained). As can be seen, walls O12_150_C, O12_P13_150_C, and SS1.5_150_C display, on average, the initial steepest increases in damage with the increase in cyclic displacement, with reductions in initial rigidity between 25 and 40% for a displacement of $2d_y$. The reductions in the rigidity of walls O12_150_C and O12_P13_150_C are mainly due to localized (near the fasteners) crushing of claddings (Figure 4b,d), whereas that of wall SS1.5_150_C is due to the shear cut and pull out of fasteners near the corners of the wall. For a similar value of horizontal displacement (about 20 mm, Figure 6), the damage is higher in wall O12_P13_150_C than in wall O12_150_C; this is due to the damage of the plasterboard in the former.

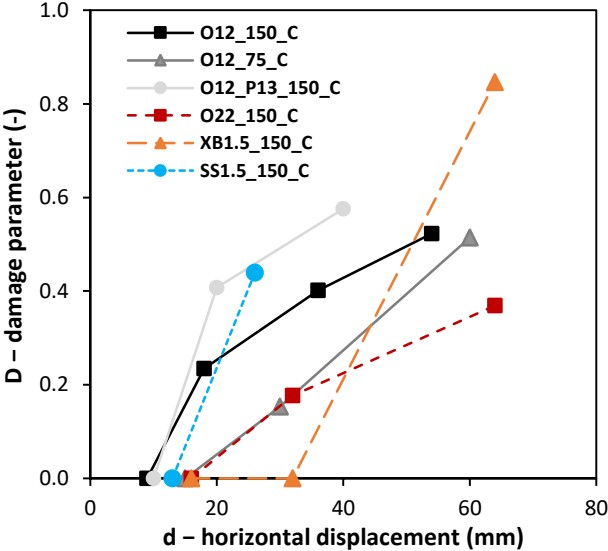

**Figure 6.** Evolution of the damage parameter, D, with the increase in horizontal displacement, d, for the walls tested under cyclic loading.

Walls O12_75_C and O22_150_C display similar evolutions of damage with the increase in the cyclic displacement amplitudes and a less steep damage evolution than that of wall O12_150_C (which can be seen as a reference). It was verified before that these three walls have an initial rigidity that does not differ from each other by more than 10% (Table 3). Hence, for a similar value of displacement ($4d_y$ = 30 mm for wall O12_150_C vs. $2d_y$ = 30 to 32 mm for walls O12_75_C and O22_150_C), the higher contact area between the fasteners and the OSB (use of more fasteners—wall O12_75_C, use of OSB with higher thickness and fasteners with higher diameter—wall O22_150_C) probably led to a lower extent of local crushing of the cladding (compared to wall O12_150_C) and, therefore, to lower damage. For a cyclic displacement amplitude of $2d_y$, the value of D for these walls is slightly lower than 0.20. At the end of the respective tests, O12_75_C presents slightly higher damage than O22_150_C.

Wall XB1.5_150_C displays a null value of the damage for a cyclic displacement amplitude of $2d_y$, which means that up to that point, any energy that has been dissipated

by the wall was essentially still associated with the yielding of tracks and studs near the corners (it was verified before that the straps remained elastic). However, for a cyclic amplitude of $4d_y$, the wall is almost fully damaged, presenting a value of D = 0.85 which, as stated and shown before, is associated with the failure of fasteners at the strap's extremities. These elements were responsible for the rigidity of this wall, unlike the remaining, which have a cladding more continuously connected to the steel structure by smeared fasteners. In the latter, the tension field rotates as the displacement increases, which allows the walls to still have some rigidity after damage in the corners occurs; however, wall XB1.5_150_C has no alternative loading path.

## 4. Conclusions

This work presented an experimental investigation on the (monotonic and cyclic) in-plane shear behavior of light steel framing (LSF) walls. The main parameters evaluated were (i) thickness and material used as cladding (OSB, plasterboard, and steel), (ii) spacing of fasteners (150 or 75 mm), and (iii) the use of steel bracing elements. The influence of these parameters on the initial rigidity, ultimate load, failure mode, ductility, and damage of the walls was addressed. The main conclusions of this work are:

- The strengths of the walls stemming from the monotonic tests seem to be good predictors of their strengths under cyclic loading since the average ratio between the latter and the former is 1.04;
- The ultimate load of walls with non-metallic cladding is mainly influenced (i) by the number of fasteners (doubling it leads to an increase of 33% of the ultimate load) and (ii) by OSB thickness (its increase, in 80%, leads to an increase in ultimate load of 13%). The use of the plasterboard did not improve the ultimate load of the LSF walls, as this element fails before the OSB; such failure only leads to a decrease in the rigidity of the wall as the load increases;
- The less brittle walls were generally those with OSB as cladding (including that with the plasterboard) with exception of that with twice the number of fasteners. The latter presented a ductility index similar to that of the wall with x-bracing, in which the ductility of steel straps was not triggered. The most brittle wall was that with the steel sheet, which presented a ductility 50 to 75% lower than the remaining walls;
- The wall with the steel x-bracing element was the most damaged (damage parameter of 0.85) at the end of testing due to the failure of the fasteners of the steel straps; the elements responsible for its rigidity. For the remaining walls, in which the tension field can rotate as the displacement increases and have a further number of fasteners to ensure load transmission, the damage parameter presented values, at the end of the tests, between 0.40 and 0.60.

**Author Contributions:** Conceptualization, D.M.L. and N.S.; methodology, D.M.L., A.P.C.D. and N.S.; validation, D.M.L., A.P.C.D. and N.S.; formal analysis, D.M.L., A.P.C.D. and N.S.; investigation, D.M.L., A.P.C.D. and N.S.; data curation, D.M.L., A.P.C.D. and N.S.; writing—original draft preparation, D.M.L. and A.P.C.D.; writing—review and editing, A.P.C.D. and N.S. All authors have read and agreed to the published version of the manuscript.

**Funding:** This research was funded by FCT, grant numbers UIDB/50022/2020, UIDB/04625/2020, and CEECIND/00630/2018.

**Data Availability Statement:** Not applicable.

**Acknowledgments:** The second author gratefully acknowledges the financial support given by FCT in the context of the postdoctoral contract CEECIND/00630/2018. The third author also acknowledges the financial support received, as this work was supported by FCT, through IDMEC, under LAETA, project UIDB/50022/2020.

**Conflicts of Interest:** The authors declare no conflict of interest.

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
