# Peer review of "Experimental Investigation of Light Steel Framing Walls under Horizontal Loading"

_buildings, doi:10.3390/buildings13010193_

Round 1
Author Response
The authors provided detailed replies to the Reviewer's comments in the attached file.

Reviewer 2 Report
The proposed work focuses on the experimental investigation of Light Steel Framing walls under horizontal loading. It is of potential interest to Buildings journal readers.
Despite the importance of the subject addressed, this work needs many improvements to be ready for the publication in the buildings journal.
Specific points of improvement :
- Literature review section must be more developed and more improved by citing previous researches.
- The objective of this research must be more developed.
- There is a luck of subsections in the manuscript.
- Experimental program section is too long. It must be divided into two sections : Section 1: "Experimental program" and Section 2: "materials and methods".
- All used standards must be mentioned in the manuscript.
- Quality of figures must be improved.
- Results need an in depth discussions.
- Explain why the initial stiffness of the walls with non-metallic cladding is influenced not only by the thickness and stiffness of the cladding but also by the number of fasteners ?
- The conclusion section is too long. Only the main results must be indicated.
Author Response
The authors provided detailed replies to Reviewer's comments in the attached file.

Round 2
Reviewer 2 Report
The proposed work focuses on the experimental investigation of Light Steel Framing walls under horizontal loading. It is of potential interest to Buildings journal readers.
I think that the revised version of the submitted paper is well improved by considering the reviewers and editor recommendations and remarks. Indeed, I think that this paper is accepted in this form